# Synthesis of High-Efficiency, Eco-Friendly, and Synergistic Flame Retardant for Epoxy Resin

Jiaxiang Gao , Hanguang Wu *, Yang Xiao, Wenjing Ma, Fei Xu, Rui Wang and Zhiguo Zhu *

School of Materials Design and Engineering, Beijing Institute of Fashion Technology, Beijing 100029, China
* Correspondence: hanguangwu@bift.edu.cn (H.W.); clyzzg@bift.edu.cn (Z.Z.)

**Abstract:** It remains a challenge to prepare flame-retardant composites via the addition of a low content of flame retardant. In this work, a novel DOPO-functionalized reduced graphene oxide hybrid (DOPO-M-rGO) flame-retardant system was synthesized for epoxy resin (EP). The phosphorus-nitrogen-reduced graphene oxide ternary synergistic effect provided DOPO-M-rGO with high flame-resistance efficiency in EP; thus, the EP-based composite exhibited superior fire-resistant performance at extremely low DOPO-M-rGO loading. The limiting oxygen index (LOI) of the EP-based composite was increased from 25% to 32% with only 1.5 wt% DOPO-M-rGO addition, and the peak heat release rate (pHRR), total heat release (THR), and total smoke production (TSP) were significantly decreased by 55%, 30%, and 20%, respectively. In addition, as a halogen-free flame-retardant system, DOPO-M-rGO presents great application potential as an eco-friendly additive for the flame-resistance improvement of thermosetting polymer materials.

**Keywords:** flame retardant; epoxy resin; synergistic effect; high efficiency

## 1. Introduction

As a polymer material with excellent properties and low cost, epoxy resin (EP) is widely used in various fields, such as adhesives, coatings, electronics, and aerospace industries [1–8]. Nevertheless, the inherent flammability of EP might pose a serious threat to the environment, people's life and property, and severely limits its further application in automobile, household, and aerospace fields. Therefore, the flame-resistance modification for EP has attracted much attention [9–19].

Graphene oxide (GO) is one of the most promising materials as the flame retardant for EP because it presents high thermal stability and an excellent barrier effect during combustion [20–24]. In addition, taking advantage of the abundant functional groups, GO is easily doped by metal ions or chemically reduced with other molecules, leading to the formation of a variety of flame-retardant systems with further improved fire-resistant performances, in which the reduced graphene oxide (rGO) component functions as the multi-layered skeleton [25–30]. In Wang's work, a flame retardant for EP was prepared by functionalizing the rGO with a sheet-like metal-organic zinc N, N'-piperazine (bismethylene phosphonate) [31]. The addition of 5 wt% of flame retardant promoted the EP-based composite to reach UL-94 V-0 rating, and resulted in 38.7%, 30.5%, and 33.7% lowered peak heat release rate (pHRR), total heat release (THR), and total smoke production (TSP) values, respectively, than that of pure EP. Zhu synthesized an iron hexamethylenediaminetetrakis-(methylenephosphonate) (Fe-HDTMP)-rGO hybrid, and used it in EP to improve the fire resistance. Compared to the pure EP, the EP modified by 5 wt% (Fe-HDTMP)-rGO exhibited 68.2% lower TSP, 54.5% lower peak CO production rate, 66.3% lower THR, and 47.7% reduced pHRR [32]. Through grafting silane coupling agent (KH550) as a bridge to connect GO and Octa (propyl glycidyl ether) POSS, Qu synthesized an organosilane-functionalized GO (FGO) as the flame retardant of EP, and the modification of EP-based composite with only 0.7 wt% FGO could reduce pHRR, THR, and TSP values by 49.7%,

34.3%, and 41.5%, respectively [33]. Xiao produced a flame retardant for EP by using melamine and GO, which is called AGO@COF [34]. The limiting oxygen index (LOI) value of the EP-based composite increased from 24% to 25.5% when 2 wt% AGO@COF was added, and the pHRR value was reduced by 43.6%. However, the flame retardancy performances of the graphene-based flame retardants, especially the UL-94 rating, need to be further improved (see Table S1).

9,10-Dihydro-9-oxa-10-phosphaphenanthrene-10-oxide (DOPO) is an H-phosphinate compound with active hydrogen, which is also widely used as the flame retardant for EP in previous studies (see Table S2) [35–41]. However, the DOPO-based flame retardants face the challenge of low flame-retardant efficiency, which can be solved by developing the hybrid flame-retardant systems of DOPO and rGO (see Table S3) [42–47]. Through the reaction between the active P-H from DOPO and the epoxy groups on GO, DOPO can be directly grafted onto the surface of GO to develop a DOPO-functionalized rGO, which plays an effective role in improving the flame resistance of the EP-based composites [42,43]. Zhi synthesized a functionalized GO grafted by DOPO and vinyltriethoxysilane (VTMS), which obviously increased the thermal stability and flame-retardant properties of EP [44]. In Ji's work, a functionalized rGO decorated with bi-DOPO groups (f-GO) was fabricated by using a covalent modification method, which endows EP with superior fire-resistant performances [45]. The extremely low loading (1 wt%) of the f-GO increased the LOI value of the composite from 19.9% to 30.8%, and the THR and the pHRR values were decreased by 44.0% and 55.5%, respectively. Furthermore, strategies have been developed to fabricate the DOPO-functionalized rGO hybrids through grafting DOPO onto the rGO skeleton by using other components as the bridge. Feng used glycidyl methacrylate as the bridge to synthesize a DOPO-functionalized rGO hybrid (GP-DOPO), and used it as the flame retardant for EP [46]. The addition of 2 wt% GO-DOPO decreased the pHRR value of the EP-based composite by 27%, the THR value is decreased by 32%, and the TSP value is decreased by 31%. In Qian's work, DOPO was firstly reacted with VTMS to form an intermediate, which was then grafted onto rGO by using (3-isocyanatopropyl) triethoxysilane as the bridge, and obtained a product which is highly effective in reducing the fire hazards of EP [47]. Although the DOPO-functionalized rGO flame retardants present higher flame-retardant efficiency, the vertical combustion grades of most DOPO-functionalized rGO flame retardants are required to be improved to achieve V-0.

As a nitrogen-containing compound, melamine is widely used as a typical eco-friendly flame retardant for EP, and its flame-resistance effect can be further improved when used cooperatively with other phosphorus-based flame retardants [48–52]. In this work, a novel DOPO-functionalized rGO hybrid (DOPO-M-rGO) was synthesized by using the synthesized methyl vinyl dichlorosilane/melamine polymeric intermediate as the bridge, which was used as a graphene-based phosphorus/nitrogen-containing flame retardant for EP. This study provides a facile approach for creating an extraordinarily effective flame retardant for EP, and aims to investigate the synergistic flame-resistance effect of DOPO, melamine, and the rGO skeleton components of DOPO-M-rGO.

## 2. Experimental Section

### 2.1. Materials

GO was prepared according to Hummer's method [53]. Melamine, methyl vinyl dichlorosilane, 1-ethyl-3-(3-dimethylaminopropyl) carbonized diimine hydrochloride (EDCI), and 4-dimethylaminopyridine (DMAP) were provided by Shanghai Maclin Biochemical Technology Co., Ltd. (Shanghai, China). DOPO, azo-diisobutyronitrile (AIBN), EP precursor (E51), and polyamide agent (650) were provided by Shanghai Aladdin Reagent Co., Ltd. (Shanghai, China). Tetrahydrofuran (THF) was provided by Beijing Tongguang Fine Chemical Co., Ltd. (Beijing, China).

### 2.2. Synthesis of DOPO-M-rGO

DOPO-M-rGO was synthesized by a two-step method, and the corresponding synthetic route is illustrated in Figure 1. First, melamine and methyl vinyl dichlorosilane (with the mass ratio of 1:1) was reacted in the THF suspension under mild stirring for 4 h to form a polymeric intermediate bearing -NH$_2$ groups and C=C bonds (see Figure 1a). Then, DOPO was added into the obtained system, and AIBN was used as the catalyst. The mass ratio of the melamine/methyl vinyl dichlorosilane polymeric intermediate, DOPO, and AIBN was controlled at 1:1:0.005. After reaction at 70 °C for 20 h, the white solid precipitate in the system was collected, washed, and dried, which is named DOPO-M in the following text (Figure 1b). Afterward, the obtained DOPO-M sample was added into THF along with GO to make a suspension, in which the mass ratio of DOPO-M and GO was 2:5. A total of 10 wt% EDCI and 1 wt% DMAP were added into the suspension as the catalyst. After 10 h reaction at 70 °C in the N$_2$ atmosphere, the ultimate suspension was filtered to remove the impurities, and the obtained black solid was abbreviated as DOPO-M-rGO in the following text (Figure 1b), which was used as the flame retardant of EP. The schematic for the preparation of DOPO-M and DOPO-M-rGO is shown in Figure 2a.

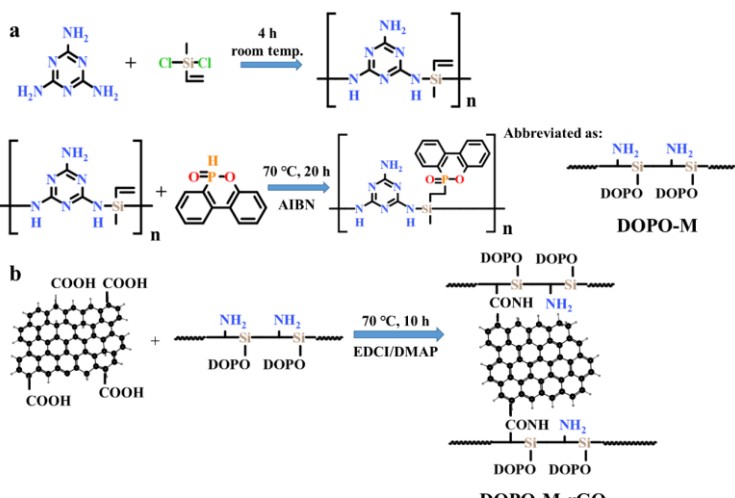

**Figure 1.** Synthetic route of (**a**) DOPO-M and (**b**) DOPO-M-rGO.

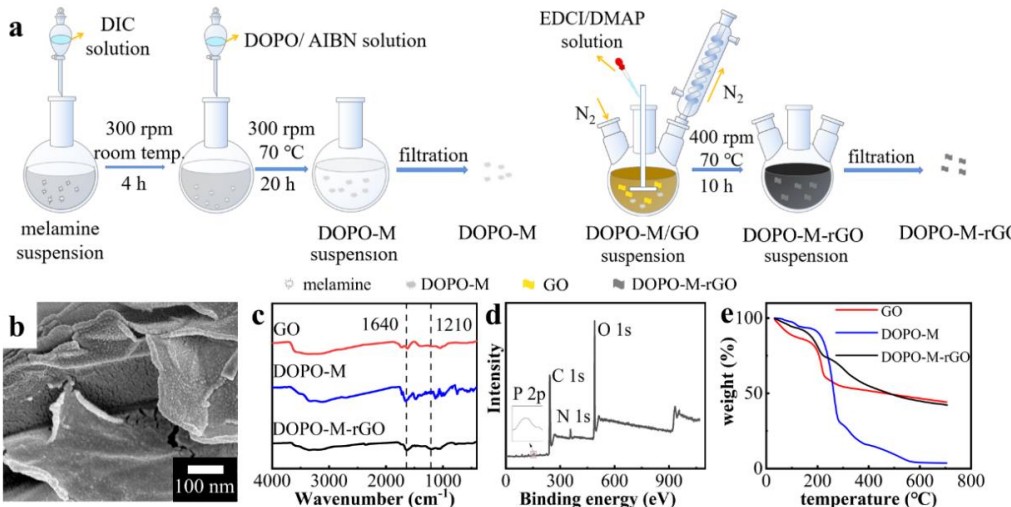

**Figure 2.** (**a**) Schematic representation for the preparation of DOPO-M-rGO; (**b**) SEM images of DOPO-M-rGO; (**c**) the FTIR spectra of GO, DOPO-M, and DOPO-M-rGO; (**d**) the XPS spectra for DOPO-M-rGO; (**e**) the TGA curves of GO, DOPO-M, and DOPO-M-rGO.

### 2.3. Preparation of DOPO-M-rGO/EP Composite

The prepared DOPO-M-rGO (0.5, 1, 1.5, 2, 3, and 5 wt%) was slowly added into the EP precursor (E51) and tenderly stirred at 60 °C for 10 min, and the polyamide agent (650) (45 wt%) was added into the system whilst continuously stirring as the curing agent of EP. Then, the obtained uniform mixture was casted and cured at 80 °C for 2 h, and the DOPO-M-rGO/EP composites with different DOPO-M-rGO contents were prepared.

The rGO/EP composite with 1.5 wt% rGO and the DOPO-M/EP composite with 1.5 wt% DOPO-M were also prepared for comparative study.

### 2.4. Characterization

Fourier transform infrared spectrometry (FTIR) was conducted by a Nicolet 6700 FTIR spectrometer (Thermo Fisher Scientific, Waltham, MA, USA) over the range of 400–4000 cm$^{-1}$. The X-ray photoelectron spectroscopy (XPS) was performed on an X-ray photoelectron spectrometer (AXIS, Kratos Analytical Ltd., Stretford, UK); the emission current and voltage were controlled at 5 mA and 10 kV, respectively. Raman spectroscopy was conducted by a SPEX-1403 laser Raman spectrometer (Renishaw in Via, London, UK) at an excitation wavelength of 532 nm. The morphologies of the samples were observed by using a scanning electron microscope (JSM-7500F, JEOL, Tokyo, Japan). The mechanical performances were measured by using an Instron universal material strength tester (4302, Instron Corporation, High Wycombe, UK) at a speed of 10 mm·min$^{-1}$ according to the ASTM D638 standard. The width of the specimen was 6.0 ± 0.1 mm and the thickness was 2.0 ± 0.1 mm. All the samples were tested five times and the average values were selected as the final results.

Thermogravimetric analysis (TGA) was used to characterize the thermal behavior of the samples. The TGA was conducted by using a TG/DTA 6300 (Seiko, Japan), and the samples were heated from 30 °C to 700 °C at a rate of 10 °C min$^{-1}$. The flame-retardant properties of the pure EP and the EP-based composite were investigated by using LOI, vertical burning testing (UL-94), and cone calorimeter test (CCT). The LOI tests were conducted on a Dynisco LOI test instrument according to ASTMD 2863-97 standard. The size of the specimen was 80 mm × 6.5 mm × 3 mm. The UL-94 tests were carried out on a vertical burning instrument (CFZ-2, Jiangning Analytical Instrument Factory Co., Ltd., Nanjing, China) according to ASTM D3801 standard. The size of the specimen was 130 mm × 13 mm × 3 mm. CCT was conducted by using a calorimeter (iCone, Fire Testing Technology Co., Ltd., East Grinstead, UK) according to the ISO 5660-1 standard under a heat flux of 35 kW·m$^{-2}$, and the sample with the dimensions of 100 mm × 100 mm × 3 mm was wrapped with aluminum foil.

## 3. Results and Discussion

### 3.1. Fabrication of DOPO-M-rGO

The fabrication process of DOPO-M-rGO is shown in Figure 2a. The chemical reaction between methyl vinyl dichlorosilane and melamine was firstly executed to synthesize a polymeric intermediate (see Figure 1a), which was then reacted with DOPO via addition reaction to form the DOPO-M. Afterward, the flame retardant DOPO-M-rGO was synthesized via the amidation reaction between the obtained DOPO-M and GO, accompanied by the reduction of GO (see Figure 1b). Therefore, by using the synthesized methyl vinyl dichlorosilane/melamine polymeric intermediate as the bridge, a DOPO-functionalized rGO hybrid flame retardant was successfully prepared, which was used as the flame retardant of EP. Figure 2b shows the SEM image of DOPO-M-rGO, in which a multilayered structure is exhibited thanks to the rGO component functioning as the skeleton. In addition, abundant clusters are attached on the rGO layers, which are formed by the grafting of DOPO-M.

The FTIR spectra of the melamine, DOPO, and GO monomers, as well as the synthesized DOPO-M and DOPO-M-rGO products are shown in Figures 2c and S1. It can be seen from the spectra of DOPO, melamine, and DOPO-M (as shown in Figure S1) that the peak at 835 cm$^{-1}$ assigned to the Si-N bond shows up in the spectrum of DOPO-M, but not in the spectrum of DOPO or melamine, indicating the successful chemical reaction between

methyl vinyl dichlorosilane and melamine and the formation of the Si-N bond in the methyl vinyl dichlorosilane/melamine polymeric intermediate. In addition, the absorption peak at 2440 cm$^{-1}$ attributed to the P-H bond of DOPO disappears in the spectrum of DOPO-M, indicating the subsequent reaction between the methyl vinyl dichlorosilane/melamine intermediate oligomer and DOPO. Therefore, DOPO-M is successfully synthesized in our study. The peaks at 1640 cm$^{-1}$ and 1210 cm$^{-1}$ assigned to the amide bond show up in the spectrum of DOPO-M-rGO, indicating that the successful amidation reaction between DOPO-M and GO, and the successful synthesis of DOPO-M-rGO. In the XPS spectrum of synthesized DOPO-M-rGO (Figures 2d and S2), the peaks at 155.2 eV, 284.5 eV, 365.7 eV, and 490.1 eV are associated with the P atoms, C atoms, N atoms, and O atoms, respectively. In the high-resolution XPS spectrum for C 1s (Figure S2a), the main peaks centered at 284.3 eV and 286.5 eV are attributed to the C-C and C-O-C/C-O-P, whereas the additional component centered at 288.2 eV and 285.7 eV are assigned to C=O and C-N, respectively. The peaks in the XPS spectrum for N 1s (Figure S2b) are attributed to N-Si (398.5 eV), C-N-C (397.9 eV), and N-H (395.7 eV), respectively. In addition, the XPS spectrum for Si 2p (Figure S2c) is deconvoluted into two peaks at 101.9 eV and 101.3 eV, which correspond to Si-C and Si-N, respectively. The XPS results indicate the formation of the Si-N bonds and the amide bonds in DOPO-M-rGO, which is inconsistent with the FTIR results.

The thermal degradation behaviors of the GO, DOPO-M, and DOPO-M-rGO are shown by their TGA curves (see Figure 2e), in which the temperature of the maximum weight loss rate can be obtained. It can be observed that the thermal degradation of DOPO-M-rGO mainly happens at a wide temperature range, which combines the thermal degradation temperature ranges of GO and DOPO-M, further indicating the simultaneous existing of GO and DOPO-M segments in DOPO-M-rGO. In addition, the residual weight of DOPO-M-rGO is much higher than DOPO-M, indicating the promotion effect of rGO in DOPO-M-rGO on the char formation during thermal degradation.

### 3.2. Fabrication of DOPO-M-rGO/EP Composite

The synthesized DOPO-M-rGO was used as flame retardant in EP, and the fabrication process of the DOPO-M-rGO/EP composite is shown in Figure 3a. The DOPO-M-rGO/EP composites were fabricated through the crosslinking of the EP monomer in a DOPO-M-rGO/EP suspension. In order to study the effect of DOPO-M-rGO content on the performances of the EP-based composite and then optimize the amount of DOPO-M-rGO as the flame retardant, different amounts of DOPO-M-rGO (0.5 wt%, 1.0 wt%, 1.5 wt%, 2.0 wt%, 3.0 wt%, 5.0 wt%) were added to form the DOPO-M-rGO/EP composites for the subsequent characterization. Figure 3b shows the stress–strain curves of the pure EP and the DOPO-M-rGO/EP composites with different DOPO-M-rGO contents, demonstrating the improvement effect of DOPO-M-rGO content on the mechanical performances of EP composites. It can be seen that the addition of DOPO-M-rGO/EP improves the tensile strength and modulus of EP when the DOPO-M-rGO content is below 2 wt%, which is mainly attributed to the good dispersion of a small content of DOPO-M-rGO and the integrity of the internal structure of the composite (see Figures 3c and S4a–d). When the amount of the added DOPO-M-rGO is above 3 wt%, the DOPO-M-rGO shows poor dispersion and a great deal of agglomeration in the composites (see Figure S4e,f), leading to the significantly deteriorated mechanical performance of the composites. Therefore, the DOPO-M-rGO content should be controlled at below 2 wt% to satisfy the practical application requirements of the composite. In addition, the DOPO-M-rGO/EP composite shows a higher modulus compared with the DOPO-M/EP composite (see Figure S3), which is mainly caused by the supporting effect of the rGO skeleton in DOPO-M-rGO.

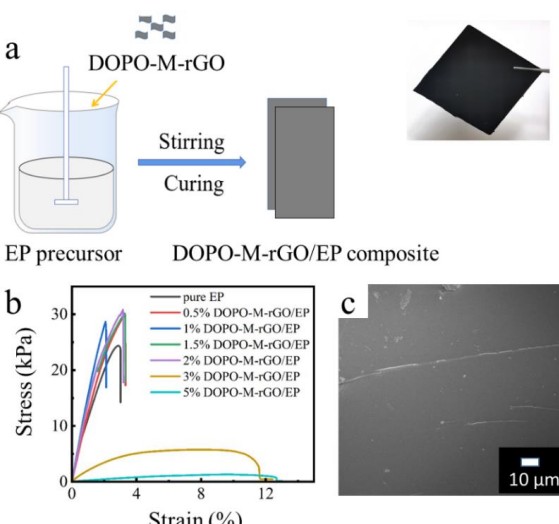

**Figure 3.** (**a**) Schematic representation for the preparation of DOPO-M-rGO/EP composite and the digital photographs of 1.5% DOPO-M-rGO/EP; (**b**) stress–stain curves of pure EP and the DOPO-M-rGO/EP composites with different DOPO-M-rGO contents; (**c**) SEM images of 1.5% DOPO-M-rGO/EP composite.

### *3.3. Flame-Resistant Effect of DOPO-M-rGO on EP*

The flame-retardant properties of the pure EP and the EP-based composites were investigated by using LOI, UL-94, and CCT.

For the pure EP, the LOI value is 25%, and it could not pass any grade in the UL-94 test. The addition of DOPO-M-rGO endows the EP-based composite with a significant improved LOI value (see Figure 4). When 1.5 wt% DOPO-M-rGO is added into the EP, the LOI value is increased from 25% to 32%, while 1.5 wt% rGO or 1.5 wt% DOPO-M has no effect on the LOI value of the EP-based composite, indicating that DOPO-M and rGO components in DOPO-M-rGO provide a synergistic flame-resistance effect on EP. In addition, when the addition of DOPO-M-rGO is increased to 1.5 wt%, the DOPO-M-rGO/EP composite reaches V-0 rating in the UL-94 test (see Table S4). Figure 5 shows the digital photographs of the pure EP, rGO/EP composite, DOPO-M/EP composite, and DOPO-M-rGO/EP composite. It can be observed that pure EP, rGO/EP composite, and DOPO-M/EP composite is easily ignited with fast flame propagation, and then fiercely burn up to the clamp within just 50 s. Actually, although the incorporation of rGO or DOPO-M into EP matrix displays no rating in the UL-94 tests, both the rGO/EP composite and DOPO-M/EP composite show a lengthened combustion time compared with the pure EP. These results demonstrate that rGO and DOPO-M can reduce the combustion speed of the EP composites, and thus endow the composites with better flame retardancy. The addition of 1.5% DOPO-M-rGO makes the EP composite reach V-0 level in the UL-94 test. The flame extinguishes itself within 5 s, which reflects the high flame retardancy of DOPO-M-rGO in EP.

CCT is widely used to evaluate the combustion performances of polymers under a forced-flaming fire scenario. In this study, we used CCT to record the thermal parameters (including HRR, THR, TSP values, and char residue) of the EP-based composite samples to assess their fire and smoke risk, and the detailed data are shown in Table 1 and Figure 6. The DOPO-M-rGO/EP composite shows the optimized flame-retardant and smoke-suppression performances when 1.5 wt% DOPO-M-rGO is added (see Figure S5 and Table S5). When the addition of DOPO-M-rGO is further increased to 2 wt%, the pHRR value, THR value, and TSP value are all significantly enhanced, which should be attributed to the incombustible gas-inhibiting effect of the hyperdense char residue formed by the DOPO-M-rGO/EP composite with high flame-retardant content [41]. In addition, the LOI value reached the highest with the addition of 1.5% DOPO-M-rGO, and the UL-94 rating reached V-0 as well.

Therefore, the DOPO-M-rGO/EP composite material with 1.5% DOPO-M-rGO was taken as the example to illustrate the flame-retardant effect of DOPO-M-rGO.

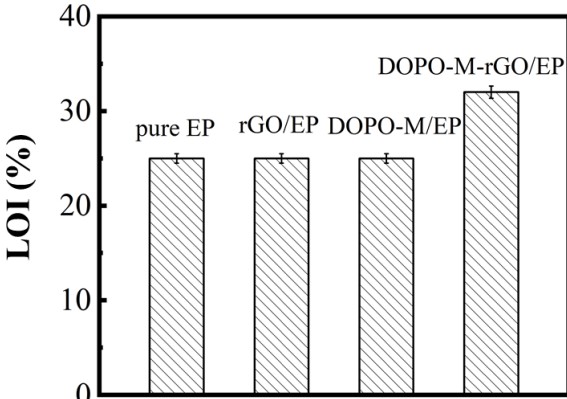

**Figure 4.** LOI values of the pure EP, rGO/EP composite, DOPO-M/EP composite, and DOPO-M-rGO/EP composite (the mass fractions of additive in the composites are all 1.5%).

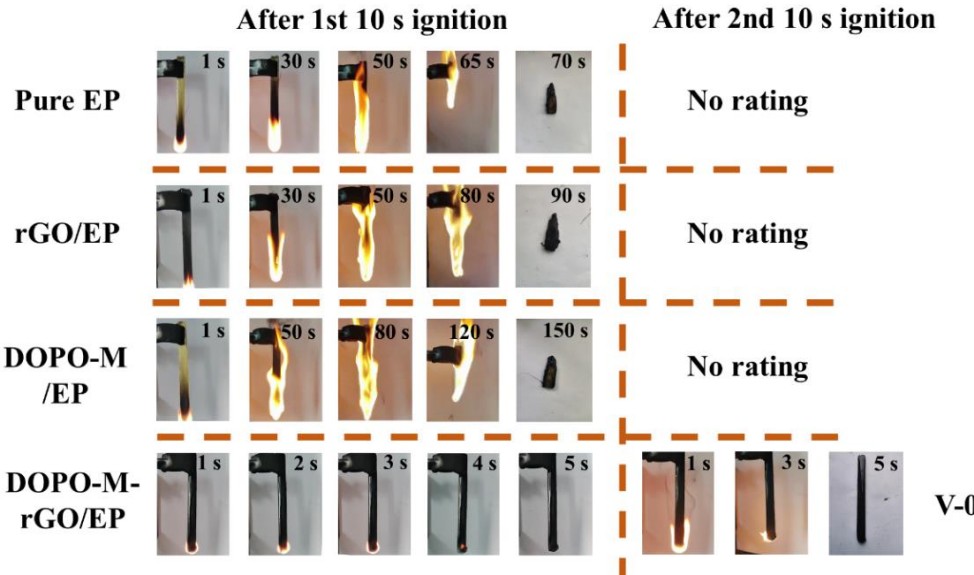

**Figure 5.** Digital photographs of the pure EP, rGO/EP composite, DOPO-M/EP composite, and DOPO-M-rGO/EP composite (the mass fractions of additive in the composites are all 1.5%) in the UL-94 test.

**Table 1.** Cone calorimeter data of the neat EP and the EP-based composites.

| Sample | Pure EP | 1.5% rGO/EP | 1.5% DOPO-M/EP | 1.5% DOPO-M-rGO/EP |
|---|---|---|---|---|
| pHRR (kW/m$^2$) | 1593 | 991 | 839 | 719 |
| Time to pHRR (s) | 156 | 167 | 162 | 141 |
| THR (MJ/m$^2$) | 125 | 130 | 99 | 88 |
| TSP (m$^2$/kg) | 65 | 45 | 74 | 52 |

The HRR curve and THR curve of the DOPO-M-rGO/EP composite with 1.5% DOPO-M-rGO are shown in Figure 6a,b, respectively, which are compared with those of pure EP, rGO/EP composite, and DOPO-M/EP composite with the same additives content. It can be seen that the pure EP burns at 100 s, showing a single pHRR of 1593 kW/m$^2$ at 156 s, and the THR is up to 125 MJ/m$^2$. The pHRR for the composite containing 1.5 wt% rGO is reduced by 38% to 991 kW/m$^2$, while the THR shows no significant change compared

with the pure EP. When 1.5 wt% DOPO-M is added into the EP, not only is the pHRR decreased to 839 kW/m$^2$, but also the THR is significantly reduced by 21% to 99 MJ/m$^2$ compared with that of pure EP. The incorporation of DOPO-M-rGO results in the further decreased pHRR and THR values of the EP-based composite (as shown in Table 1 and Figure 6). The 1.5 wt% DOPO-M-rGO addition decreases the pHRR value to 719 kW/m$^2$, and the THR value is decreased to 88 MJ/m$^2$. Therefore, as the synthesis product of GO and DOPO-M, DOPO-M-rGO simultaneously endows the EP-based composite with slower HRR and significantly reduced THR, presenting the best flame-retardant effect on EP.

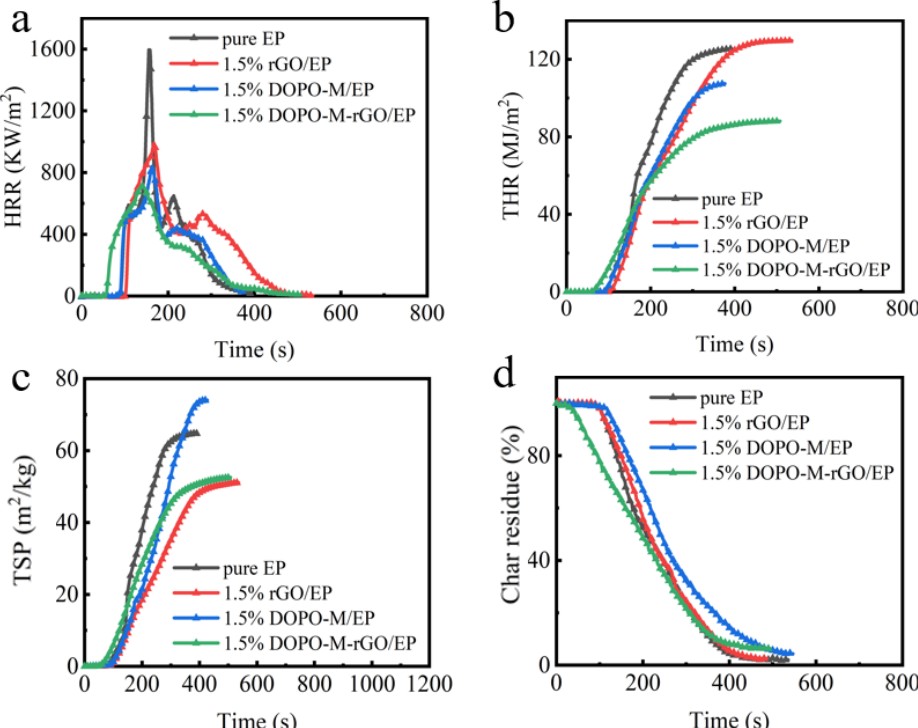

**Figure 6.** (**a**) Heat release rate; (**b**) total heat release; (**c**) total smoke production; (**d**) char residue curves of the pure EP, rGO/EP composite, DOPO-M/EP composite, and DOPO-M-rGO/EP composite (the mass fractions of additive in the composites are all 1.5%) under an external heat flux of 35 kW/m$^2$.

The smoke production of materials is regarded as one of the major factors leading to death, and the lower TSP denotes lower smoke risk and longer escaping time in fire disaster. The TSP of the pure EP and the EP-based composites are shown in Figure 6c, and the detailed data are listed in Tables 1 and S5. It can be seen that the pure EP releases 65 m$^2$/kg of TSP during the combustion, and the composite with only DOPO-M as the additive exhibits an even higher TSP value of above 70 m$^2$/kg. In contrast, the addition of 1.5 wt% rGO or 1.5 wt% DOPO-M-rGO significantly decreases the TSP values of the EP-based composites to 50 m$^2$/kg and 52 m$^2$/kg, respectively, indicating that the rGO component is the key to the smoke suppression of DOPO-M-rGO, which should be attributed to its smoke absorbing effect during the combustion [54–57]. From the char residue values of the pure EP and the EP-based composites (see Figure 6d), it can be seen that the DOPO-M/EP composite and the DOPO-M-rGO/EP composite left more char residue after the combustion than the pure EP, which should be attributed to the char layer formation promotion effect of the DOPO component in DOPO-M.

Therefore, compared with rGO and DOPO-M, the addition of 1.5 wt% DOPO-M-rGO can provide the EP-based composite with the optimal fire-resistant performances: the V-0 grade in the UL-94 test, the highest LOI value of 32%, the lowest pHRR, THR, and TSP values, indicating the excellent flame-resistant effect and smoke-suppression effect of small quantities of DOPO-M-rGO on EP. Therefore, our synthesized DOPO-

M-rGO considerably progressed in improving the flame-resistance efficiency in the EP-based composite compared with other DOPO-functionalized rGO flame-retardant systems reported previously (see Table S3).

### 3.4. Mechanism for the Flame-Resistant Effect of DOPO-M-rGO

In order to study the mechanism of the flame resistance of DOPO-M-rGO on EP, the properties and structure of char residue after CCT of the pure EP and the EP-based composites were analyzed, and the digital photos and SEM images are shown in Figure 7. It can be seen from Figure 7a,e that the pure EP almost forms no char residue, leading to the weak barrier effect. The rGO/EP composite also forms a few char fragments after the combustion (see Figure 7b,f), indicating its inferior barrier effect similar to the pure EP. Thus, the rGO/EP composite presents no obvious difference in the THR value compared with the pure EP. However, the addition of rGO can achieve a smoke-suppression effect during combustion due to its smoke absorbing ability [44–47], and thus the rGO/EP composite presents an obvious decreased TSP value (as shown in Figure 7c). The DOPO-M/EP composite forms an integrated char layer after the combustion (see Figure 7c,g), which is mainly caused by the dehydration and carbonization effect of the P-O-C and N-Si bonds in DOPO-M [58,59]. The formed thermal stabilized char layer functions as an intact shield, which effectively suppresses the transfer of heat during combustion. In addition, the incombustible gas (including $N_2$ and $NO_x$) is generated from the disintegrated melamine component during combustion, which further suppresses the spreading of flame. Therefore, the DOPO-M/EP composite exhibits a significantly declined THR value compared with EP and the rGO/EP composite. Due to the synergistic effect of the therein rGO and DOPO-M components, the DOPO-M-rGO/EP composite also forms a relative integrated char (see Figures 7d,h and S6), and the heat release and smoke release are simultaneously decreased during combustion.

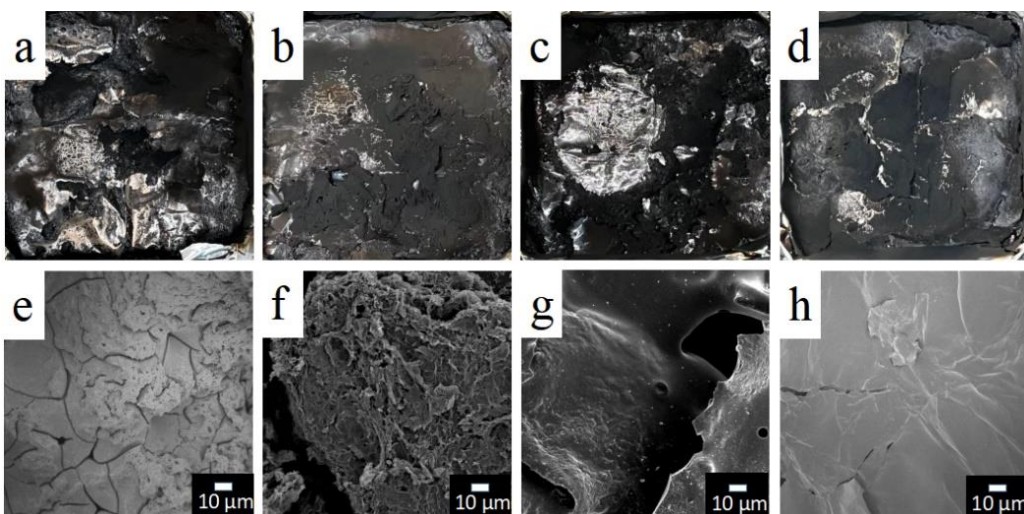

**Figure 7.** Digital images of char residues for (**a**) pure EP; (**b**) rGO/EP composite; (**c**) DOPO-M/EP composite; (**d**) DOPO-M-rGO/EP composite and SEM images of char residues for (**e**) pure EP; (**f**) rGO/EP composite; (**g**) DOPO-M/EP composite; (**h**) DOPO-M-rGO/EP composite after cone calorimeter test (the mass fractions of additive in the composites are all 1.5%).

In order to further clarify the flame-retardant mechanism of DOPO-M-rGO on EP, Raman spectroscopy of the char residues after CCT were conducted (see Figure 8). The graphitization degree of char residue can be reflected by the $I_D$ and $I_G$ ratio. The lower $I_D/I_G$ value indicates the higher graphitization degree and the higher thermal stability of the formed char residue [60]. It can be seen from Figure 8a,b that the $I_D/I_G$ value of the rGO/EP composite is 1.05, which is obviously lower than that of pure EP (1.33). This is mainly because rGO is a carbonaceous material; thus, showing an excellent graphitization effect.

For the DOPO-M/EP composites, the $I_D/I_G$ value is 1.19 (see Figure 8c), indicating that DOPO-M enhances the graphitization degree and promotes the dense structure formation of the char layer, which is well inconsistent with the SEM images in Figure 7. Under the synergistic effect of the rGO component and the DOPO-M component in DOPO-M-rGO, the DOPO-M-rGO/EP composite exhibits the lowest $I_D/I_G$ value of 1.01 (see Figure 8d).

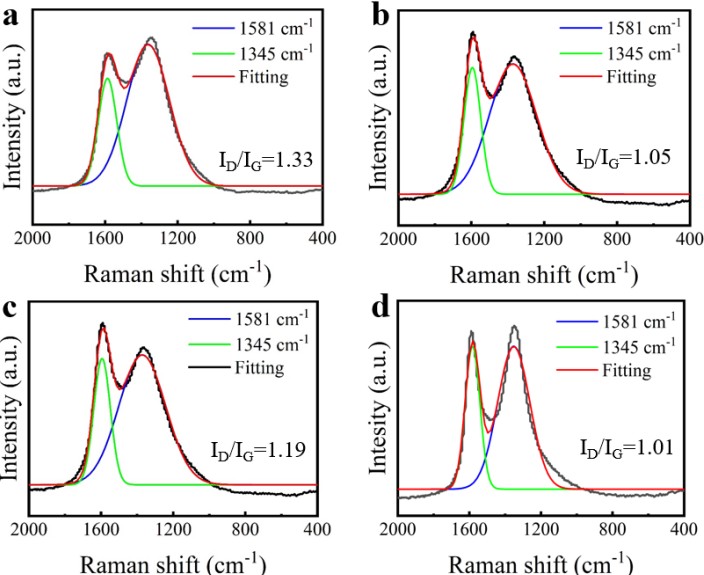

**Figure 8.** Raman spectra of char residues of (**a**) pure EP; (**b**) rGO/EP composite; (**c**) DOPO-M/EP composite; (**d**) DOPO-M-rGO/EP composite.

Furthermore, the chemical structure of char residues of the DOPO-M-rGO/EP composite was demonstrated by XPS (see Figure 9). As displayed in Figure 9a, C 1s of the char residue has peaks at 285.3 eV (C-O-C/C-O-P) and 284.4 eV (C-C). For the O 1s spectra (Figure 9b), two peaks are observed, which are attributed to the C=O/P=O (531.7 eV) or/and C-O-C (533.5 eV) groups. The P 2p peak in Figure 9c, which appears around 132.1 eV, is attributed to the P-O-C/$PO_3$ structure, which is derived from DOPO decomposition [11]. The peaks spectrum for N 1s (Figure 9d) are attributed to C-N (397.0 eV), C=N (397.8 eV), N-Si (398.5 eV), and oxidized N compounds (399.6 eV), respectively [11,61]. The result shows that P-O-C/$PO_3$ and N-Si formed through the decomposition of DOPO-M-rGO during the combustion process of DOPO-M-rGO/EP composite, simultaneously forming the char layer with high thermal stability, providing the DOPO-M-rGO/EP composite with efficient flame retardance [11].

Based on the above results, the flame-retardant mechanism for DOPO-M-rGO on EP is proposed, which is shown in Figure 10. The rGO component in DOPO-M-rGO functions as a skeleton, on which DOPO/melamine (DOPO-M) are grafted. During the combustion of the DOPO-M-rGO/EP composite, the incombustible gases, such as $N_2$ and $NO_x$, are firstly generated from the disintegrated melamine component in DOPO-M-rGO, which diluted the surrounded oxygen concentration; thus, significantly decreasing the pHRR value of the composite. With the further combustion of the composite, the decomposed DOPO-M component in DOPO-M-rGO enhances the graphitization degree and promotes the formation of a dense phosphorus-containing char layer, which presents a barrier effect and effectively suppresses the release of the heat. In addition, the rGO component in DOPO-M-rGO adsorbs the generated smoke during the combustion process to achieve the smoke-suppression effect. Therefore, rGO and DOPO-M components both play important roles in the high fire-resistant performances of the DOPO-M-rGO/EP composite. Due to its well dispersion in the EP matrix as the additive, DOPO-M-rGO in the composite presents high flame-resistance efficiency. In other words, only a small content (1.5 wt%) of DOPO-M-rGO can endow the EP-based composite with high flame-resistance performances.

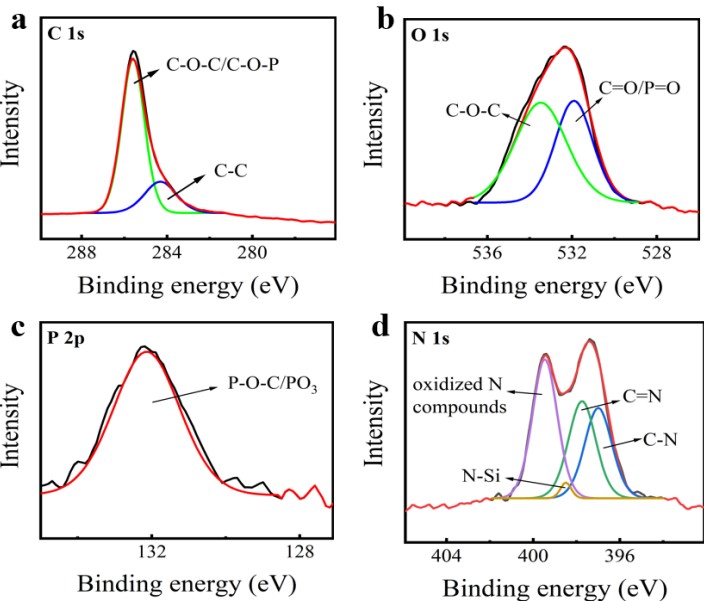

**Figure 9.** High resolution XPS spectra for (**a**) C 1s; (**b**) O 1s; (**c**) P 2p; and (**d**) N 1s of the char residue of the DOPO-M-rGO/EP composite.

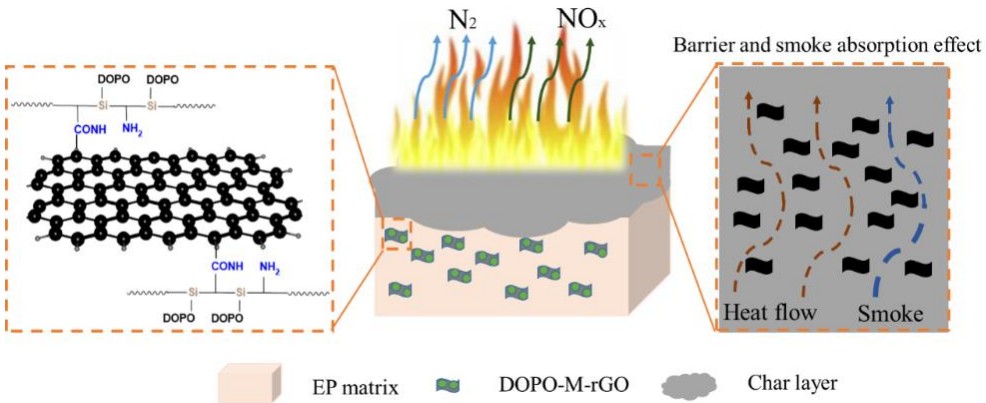

**Figure 10.** Schematic illustration of flame-retardant mechanism of DOPO-M-rGO in EP.

## 4. Conclusions

This study demonstrates a green and facile synthesis method for a novel flame retardant via a reduction of GO by the synthetic product of DOPO and melamine, which is thus regarded as a DOPO-functionalized rGO hybrid (DOPO-M-rGO). The EP-based composite with the addition of DOPO-M-rGO exhibited superior fire-resistant performance with extremely low loading of the flame retardant, which is attributed to the high dispersive and flame-resistance efficiency of DOPO-M-rGO. The DOPO-M-rGO/EP composite exhibits greatly reduced pHRR (decreased by 55%) and THR (decreased by 30%) values with only 1.5 wt% DOPO-M-rGO addition, and the LOI value is increased from 25% to 32%. In addition, the smoke production is also significantly decreased, and the TSP value is declined by 20%. The outstanding flame-retardant effect of DOPO-M-rGO in the EP-based composite is provided by the synergistic effect of the melamine, DOPO, and GO components, which promote the generation of the incombustible gases and the formation of a dense phosphorus-containing char layer during combustion. The char layer presents a barrier effect, effectively suppressing the heat release and absorbing the smoke.

**Supplementary Materials:** The following supporting information can be downloaded at: https://www.mdpi.com/article/10.3390/fire6010014/s1.

**Author Contributions:** Conceptualization, Z.Z.; data curation, Y.X., W.M. and J.G.; formal analysis, J.G., F.X. and H.W.; funding acquisition, Z.Z., H.W. and R.W.; investigation, J.G.; methodology, Z.Z. and H.W.; resources, J.G.; software, J.G.; supervision, Z.Z. and H.W.; writing—original draft preparation, J.G.; writing—review and editing, H.W. All authors have read and agreed to the published version of the manuscript.

**Funding:** This work was supported by the Joint Research Fund for Overseas Chinese Hong Kong and Macao Young Scholars (51929301), the Young Elite Scientist Sponsorship Program by BAST (BYESS2022211), and the National Natural Science Foundation of China (51903006). The Beijing Scholar Program (RCQJ20303) and Postgraduate Education Quality Improving Program in BIFT (120301990132) should also be acknowledged.

**Conflicts of Interest:** None of the authors have any financial or scientific conflicts of interest with regard to the research described in this manuscript.

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
