# Peer review of "Synthesis of High-Efficiency, Eco-Friendly, and Synergistic Flame Retardant for Epoxy Resin"

_fire, doi:10.3390/fire6010014_

Round 1
Reviewer 1 Report
In this work, a novel DOPO-functionalized reduced graphene oxide hybrid 9
(DOPO-M-rGO) was prepared and the flame retardancy of DOPO-M-rGO/EP composite was studied. The work is well organized. I suggested to do major revised.
1 As shown in abstract. “It remains a challenge to prepare flame-retardant composites via addition of low content of flame retardant.” Please compare the flame retardancy of your work with reported DOPO based flame retardant.
2 Please add more related work in introduction part, such as Chemical Engineering Journal, 2021, 405, 126946, Composites Part B 2022, 238, 109873, Chemical Engineering Journal 2021 421 127837.
3 “It can be seen that the tensile strength of the composite keeps stable with the increase in the DOPO-M-rGO content to 2 wt%, which is mainly attributed to the good dispersion of small content of DOPO-M-rGO and the integrity of the internal structure of the composite (see Fig. 3c)” I can not agree with you based on Fig. 3c. Please do more test to prove it.
4 Please add the mechanical test of pure EP, MA/EP and DOPD/EP.
5 Please add the LOI and cone test of DOPO-M/RGO/EP.
6 Fig. 1 is not clear.
Reviewer 2 Report
1. The manuscript suffers from a lack of a critical review of the current state-of-the-art and how this work reaches beyond originality and novelty. The similarity index (by iThenticate) is too high up to 43%. I think this number is not acceptable in a journal like Fire.
2. Results are not discussed based on previous work reported in the open literature. The lack of discussion makes it difficult to determine the scientific contribution of the present work.
3. The captions to the figures must give full details so that the information can be clearly understood by the reader.
4. The performance should be compared with other fire retardants and the role of GO should be further clarified.
Recommendation: Major revision
Reviewer 3 Report
Dear authors, the article and its topic, is relevant, many researchers are trying to make a fire-resistant epoxy resin and are at the beginning of the path. No one is interested in the resin itself, but in the products (structures made of it) that should be flame retardant.
There are a few comments on the article.
1. Line 206-209. Why not make the concentration with DOPO-M-rGO 2%? After all, the physical and mechanical performance remains uncritical (compared to 3%)?
2. Why didn't they make 0%? It is clear that 3% and 5% do not take. And studies at both 0% and 2% should be at the extremes of the interval.
3. Figure 3. Remove point a) as not serious, and add point c) for 0% and 2%.
4. Add photos of polymers at the same time intervals to the description of the UL-94, LOI, CCT test method results. Without comparative photos, the article looks unscientific and shown only in graphs.
5. Figure 3a and Figure 4 should be improved in terms of design (axis captions, legend size, etc.).
6. There has been a great deal of research on this subject in the USA. And several brands of flame retardant epoxy resins are known. Please make a comparison with already known flame retardant resins your results in the conclusion or in the discussion.
7. Please cite some more contemporary articles from MDPI journals on this topic.
Round 2
Reviewer 1 Report
I think it can be publised.
Reviewer 2 Report
The authors have revised the manuscript according to my comments. I would like to thank them for their efforts.
Reviewer 3 Report
The article is quite improved, but there are not enough references to European and American scientists working in the field. And there are also references to studies published in MDPI.